# What Is the Burnout of Mothers with Infants and Toddlers during the COVID-19 Pandemic? In Relation to Parenting Stress, Depression, and Parenting Efficacy

**DOI:** 10.3390/ijerph19074291

**Published:** 2022-04-03

**Authors:** Jeong-Hyo Seo, Hee-Kyung Kim

**Affiliations:** 1Department of Nursing, Graduate School, Kongju National University, Gongju 32588, Korea; kongju2018@naver.com; 2Department of Nursing, Kongju National University, Gongju 32588, Korea

**Keywords:** parenting, stress, depression, burnout, COVID-19

## Abstract

Background: The purpose of this study was to analyze the factors influencing burnout of mothers with infants or toddlers in the COVID-19 pandemic. Methods: The subjects of this study were 105 mothers who sent their children to daycare centers or kindergartens located in S and G cities. They were women who have experienced caring for children entirely at home during the COVID-19 pandemic. The collected data were analyzed using descriptive statistics, *t*-test, ANOVA, Man–Whitney U test, Pearson’s correlation coefficients, and a stepwise multiple regression using the SPSS Window 25.0 program. Results: The subjects’ burnout and parenting stress (r = 0.62, *p* < 0.001), depression (r = 0.58, *p* < 0.001), and parenting efficacy (r = −0.62, *p* < 0.001) showed a large correlation. The factors affecting the subjects’ burnout were parenting stress (β = 0.28, *p* < 0.001), parenting efficacy (β = −0.40, *p* < 0.001), depression (β = 0.27, *p* < 0.001), and spouse’s support (nearly none) (β = 0.18, *p* = 0.004). These variables explained 64.0% of the subjects’ burnout. Conclusions: Through the research results, it was confirmed that parenting stress, parenting efficacy, depression, and spouse’s support influence the mother’s burnout. Therefore, in future studies, it is necessary to expand mental health programs to lower parenting stress and depression into interventional studies on specific educational strategies such as programs to promote efficacy and improve spouse’s support.

## 1. Introduction

As COVID-19 spreads around the world, Korea is preemptively implementing strong social distancing to prevent the spread of the disease, prohibiting face-to-face education at childcare and educational institutions that accommodate the population before the school year, such as daycare centers and kindergartens, and implementing a policy to encourage parenting at home [1,2]. Even before the COVID-19 situation, housewives and working mothers did not spend all day for caring their children, so this policy creates a panic situation for mothers with toddlers and puts a burden on both housewives and working moms to look after their children all day. Therefore, though the closure of daycare centers and kindergartens is a timely and important government measure, families that need to raise and protect their children feel burdened by the lack of care for work–home balance and childcare [2]. Raising children brings great joy to parents, but it also brings great burden and stress [3]. In particular, in all societies, including Korean society, the mother’s the primary person in charge of child-rearing. During the COVID-19 period, mothers spent more time with their children at home than before, experienced care deficits during holidays and closures of kindergartens, and the role of child-rearing at home increased. The time taken to care for children increased to 6 h and 47 min after COVID-19 compared to 5 h and 3 min before COVID-19, increasing the burden of care. Moreover, the gradual easing of social distancing did not reduce stress [1,2]. In particular, working moms, who belong to dual-income couples, also spent an average of 1 h and 44 min a day caring for their children during the COVID-19 situation compared to before [2]. In addition, as the difference between work and parenting became ambiguous, the pressure to increase the productivity of work and housework was great, and they complained that it was much more difficult than before COVID-19 [4].

Moreover, in our daily life, mothers with infants and toddlers recognize that parenting is essential and important, but compared to before COVID-19, they come to recognize that it is more important after the crisis of parenting [2]. Due to the unexpectedly prolonged COVID-19 pandemic, the burden of child-rearing has been prolonged in a situation where school attendance at daycare centers and kindergartens is not free, the difference in the roles of couples became ambiguous due to a sense of fatigue and fear of infection [5]. For working moms, there was an increase in the absolute time for child-rearing after COVID-19 compared to before, and housewives also felt that they were exhausted from the stress of parenting due to the increase of the absolute time for childcare and the increase in the share of care [2]. Parenting stress experienced by mothers of young children increases the risk of child abuse at home, and social distancing can lead to problems that may make it difficult to detect and intervene in abuse at home [6].

According to previous studies, the higher the parenting stress of mothers with young children, the higher the depression, and the accumulation of parenting stress can cause problems in marital relationships and negatively affect children’s emotional stability [7,8]. In addition, mothers who experience high parenting stress show authoritarian, coercive, or rejecting attitudes toward their children, feeling atrophied, skeptical, and severely depressed about their parental abilities [9]. In addition, when the mother fails to relieve the parenting stress and depression, and is severely affected, burnout occurs. Burnout is the state where energy is not recharged and is physically and emotionally depleted due to a response to emotional stress, and a sense of efficacy is reduced and incapacitated [10]. A mother’s burnout affects herself, the child, and the whole family, aggravating confusion and becoming a social problem. This burnout is especially common when the mother feels that she can no longer recover energy for difficult tasks while the number of situations that cannot be dealt with increases during raising children [3]. Parenting stress of working moms with young children during the COVID-19 period showed a positive relationship with depression, and depression showed a positive correlation with burnout [11]. The burnout of parents, including mothers of disabled children over 3 years old, also showed a positive correlation with parenting stress and a negative correlation with parenting efficacy [3]. The burnout increases in the situation where parenting is difficult, while parenting efficacy can lower burnout.

Parenting efficacy had a moderating effect [12] of variables that mediate the process of social support, stress, and depression affecting mothers’ parenting behavior [13]. In addition, when people around the mother support and praise the mother emotionally, it strengthens the mother’s inner sense of efficacy, which in turn helps to raise children [13]. The support of the people around the mother, that is, the spouse’s support among wider social support, was a great help in reducing the mother’s parenting stress and burnout caused by COVID-19 [5]. In addition, the spouse’s support created a significant cooperative relationship in raising the son, helping solve problems, and relieved the burden of parenting by increasing the mother’s parenting efficacy of the daughter [14]. Therefore, the researcher considered parenting efficacy and spouse’s support as variables to reduce burnout of mothers with infants and toddlers even in unexpected difficult situations such as during the COVID-19 period.

Some studies have reported that mothers with infants and toddlers have parenting stress during COVID-19, but there have been only fragmentary studies on parenting stress, depression, and parenting efficacy. However, we still have insufficient studies on the relationship and influence among burnout degrees and variables. Therefore, the author intends to provide basic data for developing a nursing intervention program to lower the burnout of mothers by understanding the effects of parenting stress, depression, and parenting efficacy on the burnout of mothers caring for infants and toddlers during COVID-19 pandemic.

The purpose of this study is to analyze the effects of parenting stress, depression, and parenting efficacy on burnout in the COVID-19 pandemic and to provide basic data necessary to reduce maternal burnout in emergencies where new infectious diseases such as COVID-19 occur. The specific objectives are as follows:What is the parenting stress, depression, parenting efficacy, and burnout of mothers with infants and toddlers?What is the difference in burnout according to the general characteristics of mothers with infants and toddlers?What is the correlation between parenting stress, depression, parenting efficacy, and burnout in mothers with infants and toddlers?What factors affect the burnout of mothers with infants and toddlers?

## 2. Materials and Methods

### 2.1. Participants

The subjects of this study were 105 mothers who were sending their children aged 1 to 5 to 1 daycare center located in S city, and 1 daycare center and 1 kindergarten located in G city. The daycare center in S city is a qualitatively excellent national and public daycare center that takes care of about 65 children aged 3 to 6. In addition, the daycare center in G city is a private daycare center with 43 infants and six teachers. 66.0% of teachers are long-term workers of about 4–8 years. The kindergarten is a national kindergarten affiliated with a university and is an excellent kindergarten attended by 99 infants aged 3 to 5. There are about 11 full-time and part-time teachers. The study subjects are mothers who live in nearby apartments or houses and send infants and toddlers to daycare centers and kindergartens.

They were mothers who have experienced raising infants and toddlers entirely at home due to social distancing and the government’s policy to prevent the spread of infection during the COVID-19 period. They understood the purpose of this study and voluntarily expressed their intention to participate in writing. In the case of the number of subjects required for this study, as a result of regression analysis using the G-power 3.1.9.4 program, the effect size 0.15, significance level 0.05, and power 0.85 were maintained, then the minimum number of subjects required for analysis by inputting 5 predictors was 102. Therefore, 110 subjects were surveyed, considering a 10% dropout rate. A total of 105 questionnaires were included in the final analysis, excluding 5 copies in which the answers were insufficient.

### 2.2. Procedures

Data for this study were collected from 15 October to 30 October 2021. The author directly visited two daycare centers and two kindergartens located in S and G cities, explained the purpose and method of the research in detail to the principals, and then obtained permission. In the case of mothers sending their children to a daycare center, the author delivered the required number of questionnaires to the principal and teachers. The author collected the completed questionnaires through the principals. In the case of mothers sending their children to kindergarten, the author informed the mothers about the purpose of this study and asked them to fill out the questionnaire while waiting for the bus at the bus stop, then delivered the questionnaire in an envelope if they agreed and met them again at the bus stop the next day for collection. In addition, the homeroom teacher sent a text message about the questionnaire to all mothers, sent the questionnaires to the consenting mothers through their children, and collected the completed questionnaires. The guide for this study stated that confidentiality of personal information was guaranteed, that the questionnaire data would not be used for any other purpose, and that the questionnaire could be freely stopped. It took about 10 min to fill out the questionnaire, and a souvenir was provided to the subjects.

### 2.3. Measures

#### 2.3.1. Parenting Stress

The author used a tool modified by Choi [15] of the parenting stress tool developed by Kim and Kang [16] in consideration of the socio-cultural background of Korea to measure parenting stress. This tool consists of 21 questions in total and 2 sub-factors. The sub-factors consisted of daily stress (10 questions), burden on the role of parents, and distress (11 questions). The tool stated that it measures parenting stress after COVID-19 and asked what the subject’s level of parenting stress was compared to before COVID-19. The tool has a 5-point Likert scale ranging from 1 for “not stressful at all” to 5 for “very stressed”. A higher score indicates a higher level of parenting stress in the COVID-19 situation than before. In the study of Choi [15], the reliability of Cronbach’s α = 0.94 while the reliability of this study was 0.90. Among the sub-areas, the reliability of daily stress was 0.81, and the reliability of burden and distress for parental roles was 0.86.

#### 2.3.2. Depression

The author used the Korean version of the Depression Scale that An et al. [17] modified from the PHQ-9 (Patient Health Questionnaire-9) of Koenke et al. [18]. The tool stated that it measures the degree of depression after COVID-19 and asked how the subject’s degree of depression was compared to before COVID-19. The tool has a 9-item Likert scale ranging from 0 (not at all depressed) to 4 (very depressed). For each question, 0 points were given for “not at all”, 1 point for “not so much”, 2 points for “normal”, 3 points for “slightly yes”, and 4 points for “very yes”. A higher score indicated a higher degree of depression. In An et al. [17], the reliability of Cronbach’s α = 0.95 while the reliability of this study was 0.85.

#### 2.3.3. Parenting Efficacy

The author used the tool that Park et al. [19] modified and supplemented the Parenting Sense of Competence (PSOC) scale developed by Gibaud-Wallston and Wandersman [20] to measure parenting efficacy through factor analysis. As a result of the validity test by commissioning two experts, the two questions asking the degree of acceptance of students with disabilities did not match the children of the subject of this study, so they were deleted and a total of 27 questions were used. The five sub-factors consisted of parenting confidence (7 questions), child management ability (5 questions), health support efficacy (6 questions), capacity for growth and development (3 questions), and model ability (6 questions).

The tool stated that it measures parenting efficacy after COVID-19 and asked how much the subject’s parenting efficacy was compared to before COVID-19. The tool has a 5-point Likert scale with 1 for “No efficacy at all”, 2 for “not very effective”, 3 for “moderate”, 4 for “a little bit of efficacy” and 5 for “very high efficacy”. The higher the score, the higher the parenting efficacy. In the study of Park et al. [19], the reliability of Cronbach’s α = 0.96 while the reliability of this study was 0.94. Among the sub-areas, the reliability of parenting confidence was 0.81, the reliability of child management ability was 0.71, the reliability of health support efficacy was 0.77, the reliability of capacity for growth and development was 0.76, and the reliability of model ability was 0.74.

#### 2.3.4. Burnout

The author used the tool that Han [10] modified from the Maslach Burnout Inventory-General Survey (MBI-GS) developed for the general public by Maslach and Jackson [21] to measure the burnout of parents. The tool stated that it measures the degree of burnout after COVID-19 and asked how much the subject’s burnout was compared to before COVID-19. This tool consists of 15 questions in total and 3 sub-factors. The sub-factors consisted of exhaustion (5 questions), indifference (4 questions), and a sense of efficacy (6 questions). The six questions were coded by giving scores in reverse. The tool is on a 5-point Likert scale ranging from 1 point for “no burnout at all” to 5 points for “very severe burnout”. Negative wording was coded in reverse. The higher the score, the higher the burnout degree. In the study of Han [10], Cronbach’s α = 0.84 while the reliability of this study was 0.86. Among the sub-areas, the reliability of exhaustion was 0.85, the reliability of indifference was 0.82, and the reliability of a sense of efficacy was 0.81.

### 2.4. Statistical Analyses

The collected data were processed statistically using the SPSS/WIN 25.0 program, and the data analysis method is as follows:Descriptive statistics such as the average, standard deviation, frequency, and percentage of the subjects’ general characteristics.The subject’s parenting stress, depression, parenting efficacy, and degree of burnout were analyzed by range, average, and standard deviation.The difference in burnout according to the general characteristics of the subject was obtained by *t*-test and ANOVA, and the post-hoc test was obtained by the Scheffe test. When the assumption of normality was not satisfied, the differences between groups were analyzed using the nonparametric Man–Whitney U test.The correlation between parenting stress, parenting efficacy, depression, and burnout of the subjects was analyzed using Pearson’s correlation coefficients.Stepwise multiple regression was used to identify factors affecting the burnout of subjects.

### 2.5. Ethical Principles

This study was approved by the K University’s Institutional Review Board for the purpose, methodology, and protection of the rights of participants (KNU_IRB_2021-110). The guidelines for ethical research were followed during the study period. The consent form contained information on anonymity and confidentiality and explained that even after consenting to the research participation according to the person’s voluntary intention, she could stop participating in the research at any time and there was no disadvantage therefrom. The collected information will be managed in accordance with the Personal Information Protection Act, and the author will do her best to ensure the confidentiality of all personal information obtained through research. Informed consent was obtained from all subjects involved in the study. It was informed that the collected data will be stored for 3 years in a lockable cabinet accessible only by the author and will be discarded using a shredder after statistical analysis by computational coding for subject anonymity.

## 3. Results

### 3.1. General Characteristics of the Subjects

The subjects were mothers with infants and toddlers, a total of 105 subjects, and the age range was 27–47 years old. The average age was 36.78 ± 4.29 years, and 67.6% (71 persons) aged 30–39 accounted for the majority. Those under the age of 30 were 3.8% (4 persons) and 28.6% (30 persons) of those aged 40 or older were eligible. 54.3% (95 persons) had a religion, and 45.7% (48 persons) had no religion. In terms of education, 89.5% (94 people) had a college education or higher, and 10.5% (11 persons) finished high school. The number of children was 1–4, and the average was 1.77 ± 0.72. The number of children was 26.7% (40 persons) for one child, and 73.3% (65 persons) for two or more. Subjects having a job accounted for 60.0% (63 persons), and having no job accounted for 40.0% (42 persons). 84.8% (89 people) recognized that the support level of their spouse was higher than normal, and 15.2% (16 persons) recognized that the support level of their spouse was “almost never”. 71.4% (75 persons) had a monthly income of KRW 4 million or more, and 28.6% (30 persons) had a monthly income of less than KRW 4 million (Table 1).

### 3.2. Parenting Stress, Depression, Parenting Efficacy, and Burnout Level of the Subjects

The parenting stress level of the subjects was 2.81 ± 0.66 out of 5, among the sub-areas of parenting stress, level of burden on the role of parents, and distress was 2.90 ± 0.46. The degree of depression was 1.06 ± 0.73 out of 4. The degree of parenting efficacy was 3.66 ± 0.48 out of 5, among the sub-areas of parenting efficacy, the level of health support efficacy was 3.88 ± 0.53. And the degree of burnout was 2.28 ± 0.53 out of 5 points in full, among the sub-areas of burnout, the level of consisted of exhaustion was 2.52 ± 0.88 (Table 2).

### 3.3. Differences in Burnout According to the General Characteristics of Subjects

As a result of comparing the difference in the burnout degree according to the general characteristics of the subjects, there was a statistically significant difference in the burnout degree in spouse support (t = −3.46, *p* = 0.001) and monthly income (t = 2.09, *p* = 0.039). In other words, the level of burnout was higher in the subjects who answered that they had little support from their spouses than those who answered that they were above average. Moreover, the degree of burnout was found to be higher in subjects with a monthly income of less than KRW 4 million than those with monthly income of KRW 4 million or more. There was no statistically significant difference in the burnout degree according to other general characteristics (Table 1).

### 3.4. Correlation between Parenting Stress, Depression, Parenting Efficacy, and Burnout of Subjects

The subjects’ burnout degree, parenting stress (r = 0.62, *p* < 0.001), depression (r = 0.58, *p* < 0.001), and parenting efficacy (r = −0.62, *p* < 0.001) showed a high correlation with each other at a statistically significant level. In other words, the higher the parenting stress and depression of the subjects, the higher the burnout degree. Moreover, the lower the parenting efficacy, the higher the burnout degree. There was a correlation between depression, parenting stress (r = 0.54, *p* < 0.001), and parenting efficacy (r = −0.30, *p* = 0.002), and also a correlation between parenting efficacy and parenting stress (r = −0.40, *p* < 0.001) (Table 3).

### 3.5. Factors Affecting Subjects’ Burnout

To analyze the factors affecting the burnout of the subject, independent variables such as parenting stress, depression, and parenting efficacy were included. In addition, as a result of examining the difference with burnout according to general characteristics, variables that showed a statistically significant difference, namely spouse’s support level and monthly income, were treated as dummies and included in the analysis as control variables. As a result of verifying multicollinearity before analysis, the tolerance limits of all variables were 0.65 to 0.94, which was greater than 0.1, and the variance expansion index was 1.06 to 1.55, which was less than 10, indicating that there was no autocorrelation. To check the independence of the error term, the Durbin–Watson value was found to be 1.66 by residual analysis, which was close to 2, which satisfies the assumption of independence. Therefore, it was determined that there was no multicollinearity between the two variables, and regression analysis was performed, including all variables.

As a result of regression analysis, parenting stress (β = 0.28, *p* < 0.001), parenting efficacy (β = −0.40, *p* < 0.001), depression (β = 0.27, *p* < 0.001), and spouse’s degree of support (rarely) (β = 0.18, *p* = 0.004) were found to be a factor affecting the burnout of the subject. The explanatory power of these four variables was 64.0% (R^2^ = 0.64) and the most influential variable was parenting stress (F = 45.03, *p* < 0.001) (Table 4).

## 4. Discussion

This study analyzes the effects of parenting stress, parenting efficacy, and depression on burnout of mothers with infants and toddlers during COVID-19 and tries to provide basic data for developing burnout nursing intervention programs for mothers with infants and toddlers.

The parenting stress level of the subjects was 2.81 out of 5, which was above average. These results confirm that mothers were more stressed about raising their children due to closures after COVID-19 than before. In the pre-COVID-19 period, the parenting stress level of 182 mothers with children attending daycare and kindergarten was 2.59 [8], and the parenting stress of mothers with children attending kindergarten was also 2.59 [15]; the results of this study were higher. In addition, the parenting stress level of mothers with children attending kindergarten was 2.69 points before COVID-19 and higher thereafter [22]. It was confirmed that the gap caused a little more stress. Parenting stress is the daily stress that mothers experience in their daily life raising their children and the burden of their roles. Mothers experience physical fatigue, psychological and economic burdens due to child-rearing, and stress due to changes in new roles, tasks, and environments [23]. As a result of phenomenological studies on mothers’ parenting experiences in the COVID-19 pandemic, Mothers expressed that they suffered parenting stress in the struggle against the risk and fatigue of infectious diseases and lacked the belief that they could raise their children well [5]. In the sudden situation of COVID-19, it is necessary to implement a childcare support system such as telecommuting, family care leave, emergency care, etc. [24].

The level of depression of the subjects was not high at 1.06 out of 4 points. Before the COVID-19 period, the degree of depression of housewives and working moms with infants and toddlers was 2.39 out of 5, which was 1.91 when converted to 4 in full [8]. Moreover, the depression level of working moms with infants and children during the COVID-19 period was 1.64 out of 3, which was slightly higher than the results of this study [11]. This is different from the results of this study, and further repeated studies are needed. In general, depression is higher in women than in men, and in particular, childbirth and role performance as a mother give happiness but also induce depression [10]. The relative increase in the prevalence of depression in women is most pronounced between the ages of 24–45, and women in child-rearing are a very vulnerable group to depression. As they experience major and minor failures and losses, they sometimes fall into depression temporarily [8]. In addition to raising children, married women feel a sense of responsibility and mental and physical pressure as the role of caring for the family is increased, causing depression. Especially in times like COVID-19, daily patterns can be disrupted and lead to burnout as mothers face the problem of bearing the burden of unexpected parenting alone. Therefore, it is necessary to reduce the burden and stress of parenting, which is the cause of depression and burnout [8]. It is necessary to study depression considering women’s burden of child-rearing and the sudden situation of COVID-19.

The level of parenting efficacy of the subjects was 3.66 out of 5. This shows that even during the COVID-19 period, mothers caring for infants and toddlers were more confident than average that they could raise their children well. The results of this study were supported as it was similar to the 3.50 point of parenting efficacy of mothers of disabled children during the COVID-19 period [19]. This means that mothers who care for their children are more capable of coping when they are exposed to a sudden and challenging situation such as COVID-19. Parenting efficacy is critical in determining the quality of parenting and motivates mothers to examine their role and appropriately address issues that arise in their parenting process, enabling them to do well in their parenting behavior. In addition, mothers become receptive to their children. Providing responses and stimulation to verbal and nonverbal cues positively affects children’s adaptation, development, and social and emotional behavior [25]. Therefore, in a time of change such as COVID-19, mothers accept the problem; define the problem clearly; make a plan for problem-solving, alternative search, and implementation; and focus on problem-solving that includes self-evaluation and re-challenge. Efforts are needed to improve mothers’ parenting efficacy by developing and applying the program [23].

The burnout level of the subjects was 2.28 out of 5. Burnout is a state of physical and emotional exhaustion from efforts to satisfy the strong needs of children and families [26]. Physical, emotional, and mental fatigue due to personal depression, moral loss, decreased productivity, and cynical attitude can lead to mental health problems [10] which requires mediation intervention. Although it is difficult to interpret the results of this study due to the lack of studies that identified the burnout of the same subjects using the same tools after COVID-19, the burnout of housewives with infants and children before the COVID-19 pandemic was 2.49 points, and that of working moms was 2.58 [26]. The burnout level of mothers with children aged 3–5 was 2.28 [27], and the burnout level of mothers raising elementary school children was 2.08 [10]. Therefore, these results were similar to the results of this study, and the degree of burnout was not very high. However, considering previous studies, in the case of mothers with infants and toddlers, along with the situation before and after the COVID-19 pandemic, a systematic study is needed considering the characteristics of children and environmental situations related to parenting. Among the subjects’ general characteristics, there was a statistically significant difference in burnout in the degree of support from the spouse and monthly income, so the degree of burnout was higher in the case of almost no support from the spouse than in the case of above average support. The subjects whose monthly income was less than KRW 4 million had a higher burnout level than those with KRW 4 million or more.

The support of the spouse is an important factor in reducing the burnout of the mother. In the study of Jung [5], in the work and parenting experiences of working moms during the COVID-19 period, the spouses did not voluntarily do it first, but when they listened to specific requests for help, the mothers felt the value of their spouse and strengthened their marital relationship. In addition, the parenting stress of mothers raising infants and toddlers was lower when the parenting cooperation of the spouse was high. A mother’s physical and mental wellbeing is influenced by her spouse’s interest in and participation in child-rearing. Therefore, the results of this study are in line with the study results [27] that the spouse’s parenting cooperation can reduce the mother’s burnout by reducing the mother’s parenting burden and role burden. Above all, to increase the spouse’s support, such as participation in child-rearing, the spouse’s perception and attitude toward child-rearing must change positively. Thus, programs such as baby-rearing schools where parents participate together need to be developed and applied by the public health center and associated institutions. In addition, to increase the number of days of maternity leave for spouses and vitalize the parental leave system, education and publicity, policy preparation, system improvement, and practical strategies are needed by the state, local governments, companies, and organizations related to overcoming the low birth rate [10].

Moreover, at an economic level, in the case of working moms with young children, the low-income group had higher maternal burnout than the high-income group [10], supporting the results of this study. Economic conditions can increase the efficiency of child-rearing, so to reduce the burden of childbirth and child-rearing and reduce burnout, economic support and cost-bearing policies at national and local government level should be provided. In particular, in COVID-19, it is necessary to quickly present a strategy to solve problems by identifying problems related to child-rearing and exploring whether support and response to child-rearing are appropriate [10].

Parenting stress, parenting efficacy, depression, and burnout of the subjects showed a high correlation at a statistically significant level. In a study by Park [11], the parenting stress of working moms during the COVID-19 period was positively correlated with depression and burnout, so the higher the parenting stress and depression, the higher the burnout. Moreover, in the study of Lee [8], the higher the parenting stress of mothers with young children, the higher the degree of depression. In addition, in Choi and Koh [28], the lower the mother’s sense of parenting efficacy, the higher the burnout degree. This burnout negatively affects the mother herself and her children, so a vicious cycle of rising stress may occur. In a study on mothers of disabled children, parenting stress, parenting efficacy, and parent burnout were highly correlated, and parenting efficacy was found to have a mediating effect in the relationship between parenting stress and parent burnout [3]. Therefore, the mother’s stress causes depression, and as stress and depression continue, burnout is aggravated. Parenting stress is an important issue in that high parenting stress can cause repeated difficulties in life with children and cause annoyance or irritability, which can affect parenting behavior and eventually even affect children’s adaptation [14]. To reduce parenting stress and depression and increase parenting efficacy, it is necessary to operate a psychological program to enhance parenting efficacy. In particular, since the enhancement of parenting efficacy can have a consistent and desirable child-rearing effect and have an effect on the child’s physical, cognitive, social, and emotional development, which can have a positive effect on the child’s development [29] under stress such as COVID-19, programs that increase internal cognitive ability can be very useful coping measures.

Parenting stress, parenting efficacy, depression, and spouse’s support level (almost none) were found to be factors affecting the subject’s burnout. The explanatory power of these four variables was 64.0%, and the most influential variable was parenting stress. First, reviewing the research on parenting stress, Yoon’s [3] study also showed that parenting stress of mothers of disabled children showed a high correlation with burnout, and parenting efficacy showed a high negative correlation with burnout. During the current COVID-19 period, it is thought to be in line with the results of this study as the fact that mothers should continue to take care of their children in the absence of help from others is similar to that of parents of disabled children. Such parenting stress causes burnout, a feeling of exhaustion from one’s work, and emotional burnout in which people perceive that they are mentally overloaded [10]. In this state, they separate themselves from others, treat themselves impersonally and cynically, and evaluate themselves negatively, resulting in a lack of achievement and emotional withdrawal. In addition, the child’s temperament, the mother’s depressive tendencies, and life stress, which are factors influencing parenting stress, should be considered as a whole. It is also helpful to identify and cope with the temperament and characteristics of the child and to further enhance family interaction to reduce the mother’s depression and reduce life stress. In addition, mothers’ parenting stress can be reduced by creating a support system through the division of roles and family performance in the crisis of the COVID-19 pandemic [15].

Next, parenting efficacy was found to be a variable affecting the burnout of mothers with infants and toddlers. In a study on burnout of mothers with children aged 3–5 years, looking at the mediating effect of burnout on parenting efficacy and children’s maladaptive behavior, parenting efficacy affected burnout, and it was found that the more the child’s sense of efficacy was lacking, the higher the burnout. It was found that burnout had a partial mediating effect on children’s maladaptive behavior. If a mother with a disabled child has a high sense of parenting efficacy in raising a child, positive parenting behavior is formed, child development and growth are improved, and cognitive coping ability is improved. Therefore, in unexpected emergencies like COVID-19, parenting efficacy is inferred to foster the power to positively cope with and adapt to child-rearing [12], which can be said to be related to reducing maternal burnout. Parenting efficacy has been attracting attention as a variable mediating the process in which social resources or stress affect parenting behavior, which is a mother’s belief in the ability to raise children properly affecting the parenting behavior; or the higher the mother’s parenting efficacy, the more active participation in parenting or appropriate coping with parenting increases the efficiency of child-rearing, and thus the degree of burnout decreases [28]. Therefore, parenting efficacy is a significant factor that can lower burnout. Parenting efficacy has a significant influence on overall parenting; even if negative parenting behavior occurs due to high parenting stress, it can be positively changed if parenting efficacy is high [12]. In addition, parenting efficacy helps them easily solve problems related to child-rearing and adapt to the mother’s role [3]. Therefore, parenting efficacy acts on psychological factors such as mental burnout and can play an important role in the mother’s role, parenting behavior, parent–child relationship, and the growth and development of children [28]. Therefore, education and training to increase parenting efficacy are necessary.

In addition, depression was also found to be a factor influencing mothers’ burnout. In the COVID-19 crisis, there was a high correlation between depression and burnout of mothers with infants and toddlers [11], and depression is a major influencing factor that causes burnout due to actual dysfunction, disability in daily life, and injury [17]. Therefore, the above results were similar to those of this study, and the results of this study were supported. In a phenomenological study on the work and parenting of working moms [5], mothers are depressed in a busy state due to concerns about infection, a gap in childcare, and the burden due to the weight of parenting that they and their families have to bear. They reported suffering from feelings of regret and guilt. Women of childbearing age are very vulnerable to depression. If mothers with infants and toddlers have mental health problems such as depression, it may cause difficulties in their children’s growth and development [8]. Therefore, a mother’s mental health management program is required.

Finally, the spouse’s support was found to be a factor influencing the burnout of the subject. What was even more difficult for working moms during the COVID-19 period was a conflict with their spouses. In other words, the conflict was aggravated in that the spouses considered it the woman’s responsibility to take care of housework and nurture the children rather than to share the responsibility [24] In the results of phenomenological studies, mothers said that the most difficult thing in the COVID-19 situation was the conflict caused by the relationship with their spouse. I think this indirectly explained the importance of the spouse’s relationship and spouse’s support. The husbands of the participants took for granted that the housework and childcare were their wives’ responsibility, and they thought that they were helping, rather than sharing and taking responsibility together, making it more difficult for mothers and inducing marital conflicts. Husbands did not take the initiative to do it themselves, but they listened well when asked specifically [5]. It is thought that it can be an opportunity to increase the ease of raising children and reduce the burden by communicating with the spouse and obtaining cooperation from the spouse when requesting. Therefore, the support of a spouse is a very important variable leading to burnout following the stress and depression of mothers raising children [5]. Educational programs and workshops are needed to improve marital interaction and improve marital relationships to increase the support of spouses.

This study tried to reveal the parenting problems of mothers with infants and toddlers during the COVID-19 period. Parenting stress and parenting efficacy was higher compared to pre-COVID-19. In a disaster situation such as COVID-19, mothers with infants and toddlers had high parenting stress due to a sudden care gap, but their potential capacity in crises increased in that they had to solve the problem of parenting; so parenting efficacy increased with COVID-19. It was found that it was higher than before. This allowed mothers to recognize that their self-confidence improved when they recognized the task they had to take on as a mother. Moreover, the degree of depression and burnout did not change and was not high compared to before COVID-19. Even with these results, it is judged that depression and burnout did not appear to be high because the problem of raising children was not difficult to sustain. However, due to the parenting and care deficit caused during the COVID-19 pandemic, regardless of whether mothers were housewives or working moms, mothers with infants and toddlers suffered from poor income and work conditions related to child-rearing, resulting in a care deficit, and were unable to use daycare centers and kindergartens. The proportion of childcare at home increased, and the operation of the emergency care system was also less effective. Therefore, in preparation for disasters such as COVID-19, it is necessary to suggest measures to enhance responsiveness, such as restructuring the support system for childcare, reorganizing the emergency care system, and systemizing comprehensive childcare support services in the local community [24].

In preparation for these disasters, parenting stress, parenting efficacy, depression, and support from spouses during COVID-19 were identified as variables affecting the burnout of mothers with infants and toddlers. Therefore, a mother’s psycho-psychological management program, efficacy enhancement program, and family relationship improvement program are required to lower parenting stress and depression and increase parenting efficacy and support from spouses. This study considered various related variables to analyze the degree of burnout related to parenting during COVID-19, and based on this, helped to provide nursing interventions necessary to improve the excellence of parenting of mothers with infants and toddlers in disaster situations. The limitations of this study are as follows. This study conveniently sampled subjects from specific regions, even though it was necessary to recruit subjects representatively nationwide; this was due to the risk of infection due to the COVID-19 pandemic. Therefore, it is necessary to pay attention to generalizing the research results.

## 5. Conclusions

This study was carried out to confirm the effects of parenting stress, depression, and parenting efficacy on burnout of mothers with infants and toddlers during the COVID-19 period. It was identified that the support of the spouse was an influencing factor. This study is part of the government’s policy to prevent the spread of infectious diseases in the COVID-19 pandemic situation, and at the time of the care gap, for mothers with infants and toddlers, important concepts such as parenting stress, depression, parenting efficacy, and exhaustion for basic data of response strategies were introduced to study and derive meaningful results. Therefore, it has the advantage of carrying out meaningful and important research in that it provides a basis for a strategic plan. 

In Korea, the crisis of ultra-low birth rate is increasing in the COVID-19 pandemic. The government and local governments should provide detailed support measures to prevent a gap in care for mothers with infants and toddlers. Accordingly, support measures should be prioritized to reduce the burden of parenting on mothers and reduce burnout. In order to lower the parenting stress and depression revealed in this study and to solidify parenting efficacy and spouse support systems, it is necessary to provide various educational programs, establish administrative financial and environmental support systems, and establish strategies. In a future study, this study needs to be enlarged as an intervention study on specific educational strategies such as nursing intervention for mental health programs to reduce parenting stress and depression, and for an efficacy enhancement program to increase parenting efficacy, and communication and marital relationship improvement to increase support from spouses. In addition, based on the results of this study, it is necessary to prepare a childcare management plan and a counseling support system by nurses and teachers specializing in children in preparation for changes in parenting and overcoming difficulties in a sudden disaster such as COVID-19.

## Figures and Tables

**Table 1 ijerph-19-04291-t001:** Differences in burnout according to general characteristics.

Variables	Categories	n Mean	% SD	Mean	SD	t/F	*p*-Value
Age (year)	<30	4	3.8	2.00	0.38	0.61	0.545
	30–39	71	67.6	2.30	0.52		
	≥40	30	28.6	2.27	0.58		
		36.78	4.29				
Religion	No	48	45.7	2.31	0.55	0.44	0.661
	Yes	57	54.3	2.26	0.53		
Education	Graduate high school	11	10.5	2.11	0.49	−1.14	0.258
	Above college	94	89.5	2.30	0.54		
Number of babies	1	40	26.7	2.33	0.57	0.65	0.519
	≥2	65	73.3	2.26	0.52		
		1.77	0.72				
Job	None	42	40.0	2.35	0.58	1.13	0.261
	Have	63	60.0	2.33	0.50		
Spouse’s support *	Almost never	16	15.2	2.72	0.47	−3.46	0.001
	Above moderate	89	84.8	2.20	0.51		
Monthly income	<400	30	28.6	2.45	0.56	2.09	0.039
(KRW 10,000)	≥400	75	71.4	2.21	0.51		

* Man–Whitney U test. SD: standard deviation.

**Table 2 ijerph-19-04291-t002:** The descriptive statistics of the research variables.

Variables	Mean SD	Range
Parenting stress Daily stress Burden on the role of parents, and distress	2.81 0.66 2.71 0.68 2.90 0.76	1.29~4.29 1.20~4.20 1.09~4.45
Depression	1.06 0.73	0.00~2.89
Parenting efficacy Parenting confidence Child management ability Health support efficacy Capacity for growth and development Model ability	3.66 0.48 3.55 0.54 3.82 0.51 3.88 0.53 3.42 0.72 3.54 0.51	2.41~4.67 2.14~5.00 2.60~5.00 2.33~5.00 1.33~5.00 2.33~4.50
Burnout Consisted of exhaustion Indifference A sense of efficacy	2.28 0.53 2.52 0.88 1.71 0.66 2.46 0.58	1.13~3.87 1.00~4.40 1.00~3.75 1.17~4.00

SD: standard deviation.

**Table 3 ijerph-19-04291-t003:** Correlations among the variables.

Variables	Parenting Stress r (*p*)	Depression r (*p*)	Parenting Efficacy r (*p*)	Burnout r (*p*)
Parenting stress	1			
Depression	0.54 (<0.001)	1		
Parenting efficacy	−0.40 (<0.001)	−0.30 (0.002)	1	
Burnout	0.62 (<0.001)	0.58 (<0.001)	−0.62 (<0.001)	1

**Table 4 ijerph-19-04291-t004:** Affecting factors on the subjects’ burnout.

Variables	B	SE	β	t	*p*
Constant	3.05	0.35		8.61	<0.001
Parenting stress	0.22	0.06	0.28	3.71	<0.001
Parenting efficacy	−0.45	0.07	−0.40	−6.14	<0.001
Depression	0.20	0.05	0.27	3.81	<0.001
Spouse’s support (almost never) *	0.27	0.09	0.18	2.96	0.004

SE: standard error, * Dummy variable: spouse’s support (0 = almost never, 1 = above moderate).

## Data Availability

The data underlying this article will be shared upon reasonable request from the corresponding author.

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
