# Peer review of "What Is the Burnout of Mothers with Infants and Toddlers during the COVID-19 Pandemic? In Relation to Parenting Stress, Depression, and Parenting Efficacy"

_ijerph, 2022, doi:10.3390/ijerph19074291_

Round 1

Reviewer 1 Report

  1. It is important that no author information appears in the manuscript in order to ensure blind review. (For example: names and affiliations)
  2. In the abstract section, I recommend authors to make an effort to synthesize the results avoiding statistics data such as r; β; p)
  3. A “present study” section should be apeares instead of “Purpose”. Make sure all objectives and hypotheses are stated in this section and explain the results expected in based on previous literature.
  4. Materials and Methods section needs a reorganization and an entire revision. Now, this section is insufficiently described and not reader friendly. So, I recommend authors the following: first, delete Design section because a section of the study cannot be composed of a single sentence, which is also redundant and does not offer new information. Second, in the Statistical Analyses sections authors should be explain in details data analyses carried out in the present study. I recommend authors to rewrite this section. Third, in the Participants section authors should be explain how the sample is. Please, remove the information related to the analysis plan and the procedure and incude it in the correct section. Please describe the characteristics of the sample in more detail in terms of socioeconomic status and sociodemographic variables. I believe that include a table about the characteristics of the sample would be useful. Fourth, for the questionnaire measures that were used, please, add and ítem example of each scale or subscale.
  5. The results presented are mainly descriptive. I encourage authors to perform more in-depth/complex statistical analyzes that enrich the manuscript.
  6. In general, it would be important for the authors to work on connecting the information on the Discussion with the Introduction, integrating the interpretation of the findings. So, I encourage authors to make an effort in the narrative and support they results on a more complete literature.
  7. Please, include the limitations of the study. Moreover authors should be explain the strengths of the study.
  8. Please describe the contributions of the study more clarify in the conclusion section. At the end of the manuscript, the practical applications/implications of the study should be explained in details too.
  9. Finally, revise according to 7th Edition APA style. For example, sometimes they miss doi number, other times doi is written in the wrong way. In references:

Guy, B.; Authur, B. Academic motherhood during COVID-19: Navigating our dual roles as educators and mothers. Gender, Work & Organization. 2020; 27(5), 887-899. doi:10.111/gwao.12493

Koenke, K.; Spitzer, R.L.; Williams, J. B. The PHQ-9 validity of a brief depression severity measure. Journal of General Internal Medicine. 2001; 16(9), 606-613. https://doi.org/10.1046/j.1525-1497.2001.016009606.x

Author Response

Dear reviewer!

Thank you very much for reviewing my paper.
As you told me, I worked hard to revise it.
Thank you. I added the table of corrections after the paper reference.

Reviewer 2 Report

This Paper has scientific relevance, presented an adequate methodological design, with dense and important results for the creation of public policies that meet the needs of working women in moments of social isolation.

The results of the paper prove that there was an overload in female work and demonstrate the importance of parental support in the education of children.

Author Response

Dear reviewer !
Thank you very much for reviewing my paper.
I am attaching the last file by adding the points pointed out by other reviewer. Thank you very much.

Reviewer 3 Report

It is a well-done study that deals with a very important issue as it draws a strong focus on gender health. Further research and publications will be needed to attract more and more attention to these issues. As you underline, a crucial role will be that of training clinicians and students on the topic of gender health

Author Response

(The authors gave the same response as above.)

Reviewer 4 Report

The reviewed article concerns a very important phenomenon, which are problems in the functioning of the family and problems in parenting, in times of a COVID-19 pandemic. It is a very difficult problem and family psychologists are very concerned about it and its negative consequences. 

The article concerns the functioning of mothers during the pandemic in Korean society. In the introductory part, the authors described the specificity of mothers' functioning in this society in a very concise but interesting way. On the one hand, the presented results and concussions seem to be specific for Korean societies, but on the other hand (thanks to the tests used), they are very universal and accessible to the reader.

All the tests used, the statistical method of analyzing the results and their presentation are very correct. 

I have only one comment, a paragraf "Limitation" should be added. . In it, attention should be paid to certain limitations of the text, such as the size of the study group, which could have been important in the case of the performed statistical analyzes. 

Author Response

Dear reviewer!

Thank you very much for reviewing my paper.

As you told me, I worked hard to revise it.

Thank you so much. I added the table of corrections after the paper reference.

Round 2

Reviewer 1 Report

Let me begin by thanking the authors for their effort revising this manuscript. The manuscript is much improved as a result of these edits. I believe that now the manuscript is ready to be accepted.